

# Validation of OCO-2 error analysis using simulated retrievals

Susan S. Kulawik[1], Chris O Dell[2], Robert R. Nelson[3], Thomas E. Taylor[2]
[1]Bay Area Environmental Research Institute, Sonoma, CA, USA
[2]Cooperative Institute for Research in the Atmosphere, Colorado State University, Fort Collins, CO, USA
[3] Department of Atmospheric Science, Colorado State University, Fort Collins, Colorado, USA

*Correspondence to*: Susan S. Kulawik (susan.s.kulawik@nasa.gov)

**Abstract.** Characterization of errors and sensitivity in remotely sensed observations of greenhouse gases is necessary for their use in estimating regional-scale fluxes. We analyze 15 orbits of simulated OCO-2 with the Atmospheric Carbon Observations from Space (ACOS) retrieval, which utilizes an optimal estimation approach, to compare predicted versus actual errors in the

retrieved $CO_2$ state. We find that the non-linearity in the retrieval system results in $XCO_2$ errors of ~0.9 ppm. The predicted measurement error (resulting from radiance measurement error), about 0.2 ppm, is accurate, and an upper bound on the smoothing error (resulting from imperfect sensitivity) is not more than 0.3 ppm greater than predicted. However, the predicted $XCO_2$ interferent error (resulting from jointly retrieved parameters) is a factor of 4 larger than predicted. This results from some interferent parameter errors larger than predicted, as well as some interferent parameter errors more strongly correlated

with $XCO_2$ error than predicted. Variations in the magnitude of $CO_2$ Jacobians at different retrieved states, which vary similarly for the upper and lower partial columns, could explain the higher interferent errors. A related finding is that the error correlation within the $CO_2$ profiles is less negative than predicted, and that reducing the magnitude of the negative correlation between the upper and lower partial columns from -0.9 to -0.5 results in agreement between the predicted and actual $XCO_2$ error. We additionally study the post-processing bias correction affects errors. The bias corrected results found in the

operational OCO-2 Lite product consists of linear modification of $XCO_2$ based on specific retrieved values, such as the CO2_grad_delta (a measure of the change in the profile shape versus the prior) and dP (the retrieved surface pressure minus the prior). We find similar linear relationships between $XCO_2$ error and dP or CO2_grad_delta, but see a very complex pattern of errors throughout the entire state vector. Possibilities for mitigating biases are proposed, though additional study is needed

## 1 Introduction

OCO-2 launched in July 2014 and began providing science data in September, 2014, with the goal of estimating $CO_2$ with the "precision, resolution, and coverage needed to characterize sources and sinks of this important green-house gas." (Crisp et al., 2004). Validation of the ACOS/OCO-2 Build 7 (referred to hereafter as v7) data set (Eldering et al., 2017) versus measurements from the Total Carbon Column Network (TCCON) (Wunch et al., 2011) shows regional biases of about 0.5 ppm and standard deviations of 1.5 ppm (Wunch et al., 2017), though these errors are not entirely due to OCO-2 (TCCON and

colocation errors also contribute). Biases are particularly concerning due to propagation of $CO_2$ biases into flux biases (Basu



et al., 2013; Chevallier et al., 2014; Feng et al., 2016). OCO-2 error analysis follow Rodgers (2000) which gives a statistical estimate of errors using first-order analysis that assumes that the forward model is linear and estimates errors due to smoothing, radiance measurement error, and interferent species. The predicted $XCO_2$ errors for v7 OCO-2 are typically 0.4 ppm for ocean glint and 0.5 ppm for nadir land, which underestimate the actual errors by at least a factor of 2 (Wunch et al., 2017). The cause

of regional biases is thought to be underestimated interferent error or missing components of error analysis but is not well understood. Connor et al. (2016) found that missing physics in the forward model (e.g. more aerosol types; spectroscopy error; instrument error) leads to significantly larger posterior uncertainties than predicted by the current ACOS error analysis, using a purely linear error estimation framework. However this study finds that non-linear retrievals this relatively simple simulation system (e.g. no spectroscopic errors, no instrument noise, consistent aerosol types between the true and retrieved states) also

shows a similar relationship between predicted and actual errors, with the actual error about twice the predicted.

Cressie et al. (2016) estimates the size of second-order terms of the error analysis. The second order terms contain derivatives of the averaging kernel, gain matrix, and Jacobians with respect to state parameters. Cressie et al. (2016) estimates the errors resulting from second-order error analysis are on the order of 0.2 ppm, but this analysis was dependent on the states and sizes

of deviations used to calculate the second-order derivatives. Cressie et al. (2016) found that second order terms can cause both larger errors and biased results.

This paper explores the errors in the ”full physics” retrieval system using a simulated system with no mismatches in the retrieval versus true state vector, and no spectroscopy or instrument errors. The actual errors covariance of (retrieved minus

true) for this retrieval system are about twice the predicted errors. The linear analysis of Connor et al. (2016) does not explain the higher errors because these simulated results do not include unaccounted errors sources. Cressie et al. (2016) also does not explain the higher actual errors, because Cressie et al. (2016) estimates the second-order error as about 0.2 ppm, whereas the unaccounted error is about 0.8 ppm in this paper. In order to identify the source of the unaccounted error, actual errors are ompared to the predicted linear errors for a series of setups.

The ACOS Level 2 (L2) ”full physics” retrieval algorithm used to estimate $XCO_2$ from OCO-2 employs optimal estimation using 3 near infrared bands: (1) 0.76 μm containing significant O2 absorption ("O2 A-band"), (2) around 1.6 μm containing weak $CO_2$ absorption ("weak $CO_2$ band"), and (3) near 2.1 μm containing strong $CO_2$ absorption ("strong $CO_2$ band"). Prior to the main retrieval, a series of fast pre-processing steps are performed for quality analysis (primarily to screen out clouds)

and to provide estimates of chlorophyll fluorescence (Frankenberg, 2014). Only soundings that are deemed sufficiently clear are selected to be processed by the computationally expensive L2 retrieval. In the optimal estimation L2 retrieval used in this simulation, 45-46 retrieval parameters are simulateously estimated, including $CO_2$ volume mixing ratios at 20 pressures, albedos in 3 bands, 4 types of aerosols, meteorological parameters (temperature, water vapor, surface pressure), dispersion (frequency offset), wind speed (ocean only), and fluorescence (land only).



The retrieved $CO_2$ profile is then collapsed into a column, $XCO_2$. Recent work has alternatively partitioned the information into two partial columns (Kulawik et al., 2017). Post-processing quality screening and linear bias corrections based on various L2 retrieved parameters are then performed on $XCO_2$. The corrections are based on the slope of $XCO_2$ error versus different retrieved values, where the $XCO_2$ error is estimated from retrieved $XCO_2$ minus either (a) a constant value, in the southern

hemisphere, "The Southern Hemisphere Approximation", (b) values from surface-based observations from TCCON stations, (c) the mean of small areas (less than 1 degree), or (d) a multi-model mean (Mandrake et al., 2015). We study the effects of the post-process bias correction in Section 4.3. The simulations in this paper differ from the operational retrievel in that the fluorescence true state is set to zero, although fluorescence is still retrieved; and amplitudes of spectral residual patterns are not retrieved; except for these minor differences, these simulated retrievals are identical to the operational v7 retrievals. We

refer the interested reader to O'Dell et al (2018) for a full description of the operational retrieval, including retrieved variables and bias correction.

Simulation studies can be used to understand and probe retrieval results. There are many different ways to assess errors, listed here in order of increasing complexity and non-linearity:

1) Linear estimates of errors, which assumes linearity of the retrieval system (Connor et al., 2008; Connor et al., 2016), useful for surveying impacts of different errors with linear assumptions

2) Retrievals using a simplified radiative transfer, called the surrogate model (Hobbs et al., 2017), which incorporates non-linearity, is very fast, but does not result in the discrepancy of larger actual versus predicted

error

3) Retrievals of simulated data generated using the operational L2 forward model, called the "simplified true state", which has the advantage that the true state is within the span of the retrieval vector and the linear estimate should be valid

4) Studies using a more complex and accurate radiative transfer model to generate the observed radiances (e.g.

Raman scattering, polarization handling, surface BRDF effects) and discrepancies between the true and retrieved state vectors (e.g. aerosol type mismatches between the true and retrieval state vector, albedo shape variations) (e.g. O'Dell et al., 2012).

This paper uses system (3), which makes it easier to interpret the actual versus expected performance of the retrieval system.

System (3) was used because preliminary studies found that the performance of systems (3) and (4) were comparable. Note that the observed radiance is generated with slightly different code than the retrieval system but they are matched as closely as possible.



## 2 Retrieval system

### 2.1 Description of the OCO-2 L2 retrieval algorithm and error diagnostics

The ACOS optimal estimation approach is described in O'Dell et al. (2012, 2018) and Crisp et al. (2010). In this section we review the parameters in the retrieval vector and the equations for error estimates. The retrieved parameters for this simulation

study are shown in Table 1.

All non-$CO_2$ parameters are called interferents, and the propagation of errors from these parameters into $CO_2$ is called "interferent error".

The a priori error for $CO_2$ is a 20x20 matrix, and has strong correlations as shown in Fig. 2 of O'Dell et al. (2012). The $CO_2$ a priori error specifies 48 ppm error at the surface, 12 ppm in the mid-Troposphere, and 1.4 ppm error in the stratosphere. This is consistent with aircraft variability, but biased towards surface variability; in the ACOS retrieval, about 8% of the true mid-Tropospheric $CO_2$ variations are incorrectly attributed to surface variations based on the bias correction of CO2_grad_delta (Kulawik et al., 2017). The a priori errors for other parameters are all uncorrelated, and can be found in the L2 Product file.

The predicted errors, found in the OCO-2 L2 product as "xCO2_error_components", are based on the assumption that the non-linear, iterative retrievals can be represented as a linear estimate (Connor et al., 2008; Rodgers, 2000), and shown in Eq. 1:

$$\hat{\mathbf{v}} = \mathbf{v}_a + \mathbf{A}_{vv}(\mathbf{v}_{true} - \mathbf{v}_a) + \mathbf{A}_{ve}(\mathbf{e}_{true} - \mathbf{e}_a) + \mathbf{G}_v\boldsymbol{\varepsilon} \qquad (1)$$

where

- $\hat{\mathbf{v}}$ is the retrieved $CO_2$ profile, size $nCO_2$ (20 for OCO-2). This variable is called "$\mathbf{u}$" in Connor et al., 2008.
- $\mathbf{v_a}$ is the a priori $CO_2$ profile, size $nCO_2$
- $\mathbf{v}_{true}$ is the true $CO_2$ profile, size $nCO_2$
- $\mathbf{A}_{vv}$ is the $nCO_2$ x $nCO_2$ $CO_2$ profile averaging kernel
- $\mathbf{A}_{ve}(\mathbf{e}_{true} - \mathbf{e}_a)$ is the cross-state error representing the propagation of error from non-$CO_2$ retrieved parameters, $\mathbf{e}$ (aerosols, albedo, etc.), into retrieved $CO_2$.
- $\mathbf{e}_a$ is the a priori interferent value, size $n_{interf}$. For this work, $n_{interf}$ is 26(27) for ocean (land).
- $\mathbf{e}_{true}$ is the true interferent value, size $n_{interf}$
- $\mathbf{A}_{ve}$ is size $nCO_2$ x $n_{interf}$
- $\mathbf{G}_v$ is the gain matrix for $CO_2$, size $nCO_2$ x $n_f$, where $n_f$ is the number of spectral points, and
- $\boldsymbol{\varepsilon}$ is the spectral error, also called measurement error, size $n_f$

The full gain matrix, $\mathbf{G}$, maps from spectral signals to retrieval parameter changes, and is:

$$\mathbf{G} = (\mathbf{K^T}\boldsymbol{S_\varepsilon}^{-1}\mathbf{K} + \mathbf{Sainv})^{-1}\mathbf{K^T}\boldsymbol{S_\varepsilon}^{-1} \qquad (2)$$





where $\mathbf{K}$ is the Jacobian (or Kernel) matrix, and $\mathbf{S}_\varepsilon$ is the error covariance of the spectral error, $\varepsilon$. Note that $\mathbf{G}$ is size $n$ x $n_f$, where $n = n\text{CO}_2 + n_{\text{interf}}$ is the total number of retrieved parameters. $\mathbf{K}$ is a matrix of derivatives giving the sensitivity of the radiance at each frequency to each retrieved parameter; e.g. for the $CO_2$ parameter at 800 hPa,

$$\mathbf{K} = \frac{dRadiance}{d(CO2 \ @ \ 800 \ hPa)} \qquad (3)$$

An assumption of the ACOS retrieval system is that the Jacobians are fairly invariant during the retrieval process, as is the case in nearly all optimal estimation retrievals (see e.g. Rodgers, 2000).

10  The averaging kernel, $\mathbf{A}$, is one of the most fundamental and useful quantities in Bayesian inversion theory. It describes the predicted linear dependence of the retrieved state on the true state and prior. The diagonal of the averaging kernel gives the degrees of freedom for signal for each retrieval parameter. The averaging kernel is calculated as:

$$\mathbf{A} = \mathbf{GK} \qquad (4)$$

As will be shown in Section 3.1, we find that $\mathbf{K}_{CO2}$ varies depending on the retrieved state (indicating non-linearity), which would result in error in retrieved $CO_2$ that is not captured in the predicted errors.

After an inversion is complete, the pressure weighting function $\mathbf{h}$ (size $n\text{CO}_2$) is used to convert the retrieved $CO_2$ profile to
20  $XCO_2$ by tracking the contribution from each level to the column quantity;

$$XCO_2 = \mathbf{h}_x CO_2^{\mathrm{T}\cdot} \hat{\mathbf{v}} \qquad (5)$$

The predicted errors on the estimated $XCO_2$ arise from 3 separate terms in Eq. 1;

1. $\mathbf{G}_x \boldsymbol{\varepsilon}$ results from the errors on the measured radiances (measurement error),
2. $\mathbf{A}_{vv}(\mathbf{v}_{true} - \mathbf{v}_a)$ results from both imperfect sensitivity and constraint choices (smoothing error)
3. $\mathbf{A}_{ve}(\mathbf{e}_{true} - \mathbf{e}_a)$ results from jointly retrieved species propagated into $CO_2$ (interferent error)

30  The $CO_2$ profile can also be partitioned into a lower and upper partial column (Kulawik et al., 2017). These can be calculated using equations similar to Eq. 5, with h set for the lower partial column air mass (LMT) by zeroing out the upper 15 levels, and h set for the upper partial column (U) by zeroing out the lower 5 levels. In this work, the lower and upper partial columns are explored to try to understand the reasons behind the underpredicted $XCO_2$ errors, and the effect of the CO2_grad_delta component of the bias correction.




One important diagnostic to assess retrieval quality is the chi-squared statistic, $\chi^2$, an estimate of how well the modeled radiances match the observed radiances, defined;

$$\chi^2 = \frac{1}{n_f} \sum_f ((\text{rad\_fit}_f - \text{rad\_obs}_f) / \boldsymbol{\varepsilon}_f)^2. \qquad (6)$$

Where rad_fit is the fit radiance, rad_obs is the observed radiance, ε is the radiance error. For OCO-2, $\chi^2$ is calculated separately for each of the three OCO-2 spectral bands.

In reality, Eq. 1 would contain many additional error terms that are not considered in these simulations, e.g. spectroscopy,
instrument characteristics, aerosol mismatch errors (i.e. picking the wrong aerosol type to retrieve). These are discussed in detail in Connor et al. (2016) as linear error estimates. The results reported here only address errors in the full non-linear retrieval system for the actual retrieved variables; it does not include errors from unincluded physics or other error sources (such as spectroscopy error). In the analysis presented in Section 3, each of the diagnostics given in Equations 1 through 5 will be used to examine the error estimates on the simulations and compared to previously published results on real OCO-2 data.

**2.2 Description of the simulated dataset**

The simulated data set analyzed in this study is comprised of a set of realistic retrievals using the ACOS b3.4 version of the retrieval algorithm. It is a slightly modified version of that described in detail in O'Dell et al. (2012) (which discussed b2.9), described more fully in O'Dell et al. (2018). Table 2 shows the most important changes to the L2 retrieval algorithm between b2.9 and b3.4.

Although newer versions of the OCO-2 L2 algorithm exist (currently b8 as of time of writing), the work presented here was initially begun prior to the launch of OCO-2 in July 2014. In addition, certain tests, where the L2 true state is directly related to the retrieval vector, were simplified by using the older version of the retrieval algorithm which contains a less complicated aerosol scheme. In the older L2 algorithm versions (pre B3.5), the state vector for all soundings always included the same four
aerosol types; cloud water, cloud ice, Kahn 1 (a mixture of course and fine mode dust aerosols) and Kahn 2 (carbonaceous mode aerosols) (described more in Nelson et al. (2016)). Both Kahn 1 and 2 types contain some sulfate and sea salt aerosols as well. Newer versions of the OCO-2 L2 retrieval include a more complicated scheme in which each sounding includes water and ice, and picks the two most likely aerosol types based on a MERRA monthly climatology for the particular sounding location. The aerosol fits use a Gaussian-shaped vertical profile for each of the four types, as described in O'Dell et al. (2018).

Inputs to the b3.4 L2 retrieval algorithm include simulated L1b radiances and meteorology (taken from ECMWF) that were generated using the CSU/CIRA simulator (O'Brien et al., 2009). The simulator is driven by satellite two-line-elements which





are used to provide the satellite time and position. The code calculates relevant solar and viewing geometry and polarization, and takes surface properties from MODIS. Only a single day's worth of orbits (15 orbits on 17 June 2012) at reduced temporal sampling (1Hz instead of the operational 3Hz) and with only 1 footprint per frame (instead of the operational 8) are presented in this work. This yields approximately 2700 soundings per orbit, totaling about 40,000 soundings. Unlike real OCO-2 viewing

modes (see Crisp et al. (2017)), the simulations were generated with nadir viewing over land and glint viewing over water. Therefore no nadir-water, glint-land or target mode simulations exist as are found in real OCO-2 data. The spectral error for these simulations assumes Gaussian random noise, following the OCO-2 noise parameterization as described in Rosenberg et al. (2017).

Although the simulations do include realistic clouds and aerosols from a CALIPSO/CALIOP (Winker et al., 2010) monthly climatology, the radiative transfer portion of the simulator code allows clouds and aerosols to be switched off, making it easy to generate clear-sky radiances used in this research. The OCO-2 instrument model, described in detail in (Connor et al., 2009), was used to add realistic instrument noise to the radiances prior to running the L2 retrieval for the noise-less simulations. The operational OCO-2 dispersion and ILS values, as well as its polarization sensitivity, were used to sample the top of atmosphere

radiances. The same solar model as used in the operational retrieval was used in the L1b simulations. In addition, the A-Band Preprocessor code described in Taylor et al. (2016) was run on the cloudy-sky L1b simulations to provide realistic cloud screening prior to running the L2 retrieval.

This error analysis study ideally would use the exact same forward model in both the L1b simulations and the L2 retrieval

algorithm, as our goal is not to study errors from imperfect forward model physics, but rather errors due to physics that is perfectly described by the retrieval forward model. However, in reality these two code bases are very similar but not identical. For example, the number of vertical levels within the two code bases differ. Reasonable attempts were made to put the L1b simulations on the same footing with the L2 forward model, but minor model mismatches may remain. We do not believe these minor differences affect our primary results.

Our goal in this work is to compare linearly predicted vs. true retrieval errors, specifically in terms of three primary contributions to retrieval error discussed above: measurement, smoothing, and interferent error. Several different configurations were used to allow the estimation of the true error for each of these error components, as shown in Table 3. The "clear results" have no clouds or aerosols in the true state, however the retrieval is free to insert clouds into the retrieved state

(and given that aerosols are retrieved as ln AOD, the retrieved states is never fully aerosol-free).

Results from different configurations are intercompared to validate the individual measurement, smoothing, and retrieval errors. These predicted errors are compared to the "true" errors resulting from nonlinear retrievals, which are the retrieved minus true values.





### 2.3 Post-processing quality screening

Similar to retrievals from real observations, the simulated retrieval results need screening to remove cloudy scenes (e.g. see O Brien, 2016; Polonsky et al., 2014). Because pre-screening is not perfect, the $XCO_2$ estimates from some soundings are of low quality, even if they converge. Post-processing screening is handled through calculation of quality flags, taken from Table

5 of Polonsky et al. (2014). These flags are (a) $\chi^2 < 2$ for cases with measurement error, or $\chi^2 < 1$ for cases with no measurement error, (b) retrieved aerosol optical depth <0.2, and (c) degrees of freedom > 1.6 (degrees of freedom are defined near Eq. 4). The 3 bands are averaged to calculate the $\chi^2$ for the scene.

Table 4 shows the effects of applying post-processing quality screening for the different configurations from Table 3. The

results are separated into land and ocean scenes; approximately 1/3 pass post-processing quality screening for cloudy cases; about 80% pass post-processing quality screening for cloud-free cases. For cloudy cases, 11% and 28% of cases passing pre-screening for ocean and land, respectively, and 25% and 43% of cases passing post-screening for ocean and land, respectively. These are low compared to OCO-3 simulation studies (Eldering et al., 2018), where 25-30% of cases passed pre-screening, and 50-70% of cases passed post-screening. Some of the quality flags used for the OCO-3 studies (particularly the pre-

processing flags) are not available in our study so it is hard to directly compare throughput. The lower throughput suggests that the cloud cases or other aspects of this study were harder than the OCO-3 simulation studies.

### 2.4 Comparisons of retrieved values to true

Table 5 shows $XCO_2$ biases and errors for the different configurations from Table 3. The quantities calculated for Table 5 are the bias (the mean retrieved minus true values) and standard deviation (the square root of the moment of the retrieved minus

true difference). These quantities indicate the overall quality of the results for each configuration. The results in Table 5 are sorted by standard deviation. The worst result by far is the cloudy case with no post-processing screening. This has ~10 ppb error for land and ~3 ppm error for ocean. Ocean generally does better than land; post-processing screening generally does better than no screening; and clear cases do better than cloudy cases. The addition of measurement error has a negligible effect on standard deviation for this testing. The bold entry in Table 5 represents the most realistic "real-life" case (+measurement

error, +clouds, +post-processing screening). This has 0.8 ppm standard deviation for land and 0.7 ppm standard deviation for ocean.

In the post-processing screened data, the main concern is the -0.5 ppm bias in the clear land retrieval. We have seen this in other sets of simulations and it is an unresolved issue at this time. Recently we did find a minor bug in the simulator code that

caused a small mismatch between the water vapor profile used to calculate the L1b radiances and that written to the meteorology file that is then used in the L2 retrieval. It is possible that other minor bugs of this nature are driving the clear-sky bias, with errors mitigated by clouds in the cloudy cases.





Figure 1 shows a scatter plot of the retrieved versus true $XCO_2$ (both with the apriori subtracted). The lower panels in Figure 1 show the histogram of differences, which range from about -1.25 to +1.5 ppm for land and -1.5 to +2 ppm for water soundings. Bias correction, discussed in Section 4.3, further improves the land results by 0.1 ppm in the bias and standard

deviation as seen in Table 5, but does not improve ocean results. The standard deviation of (retrieved – true) (green dashed line) and (retrieved – linear estimate) (blue dashed line) are very similar; the linear estimate does not estimate the results any better than 0.7 to 0.9 ppm, and gives an estimate of the nonlinearity.

For real OCO-2 v7 data, comparisons to TCCON for single-observation land nadir and ocean glint show errors of 1.3 and 1.0

ppm, respectively (Kulawik et al., 2017), meaning that the real errors are 0.3-0.4 ppm larger than in these simulated data, i.e., approximately 30 to 40% larger. Real OCO-2 data has location-dependent biases on the order of 0.5 ppm (Wunch et al., 2017), compared to the much lower overall bias of 0.1-0.2 ppm seen in this simulated dataset. However, a location-dependent or seasonally dependent bias would be hard to probe with this simulated dataset, which covers only 1 day of observations.

### 3 Validation of errors and non-linearity

In this section the different error components that were introduced in Section 2.1 are isolated as much as possible to evaluate each one separately. The Averaging Kernel and Jacobians, introduced in Section 2.1 are used as diagnostics. In addition, the linearity, or lack thereof, of the system is explored.

### 3.1 System linearity

To test the system linearity, Eq. 1, the linear estimate, is compared to the non-linear retrieval. Table 6 shows the results for

cases passing post-processing quality screening, clouds, and no measurement error (Table 3, case d) using 1) the first two terms on the right side of Eq. 1 (i.e. only the $CO_2$ part of the Averaging Kernel) or 2) all of Eq. 1 (i.e. utilizing the interferent terms). The last term of Eq. 1 is not used for the noise-free case. The bottom entry in Table 6, showing retrieved vs. true $XCO_2$ (without averaging kernel applied). The comparison of retrieved $XCO_2$ versus the linear estimate have biases between 0.2 ppm and 0.9 ppm and standard deviation between 0.6 and 0.9 ppm. The bias is worse if the full averaging kernel is used.

Looking through parameter by parameter, the band 3 albedo average causes most of the large bias for the full averaging kernel for ocean. The difference between the linear estimate and the non-linear retrieval is an estimate of the non-linear error in the retrieval system.

Another way to test system linearity is to look at the consistency of the sensitivity of the system to changes in $XCO_2$, i.e., how

constant are the $XCO_2$ Jacobians (defined in Eq. 3)? For example, consider if the $XCO_2$ Jacobian weakens when an interferent, e.g. call it interferent #1, increases. If interferent #1 is larger than its true value, the $XCO_2$ Jacobian will be weaker than the





true $XCO_2$ Jacobian. If the $XCO_2$ Jacobian is weaker than the true Jacobian, then more $XCO_2$ is needed to account for the radiance differences observed, resulting in a positive bias in $XCO_2$. This would result in a positive correlation in the errors of interferent #1 and $XCO_2$. This error correlation would not be predicted by the linear error analysis because the linear error analysis assumes that the Jacobians do not vary. This could explain the stronger error correlations seen

To calculate an error resulting from varying Jacobians requires calculating second order terms, like dJacobian[$XCO_2$]/d[H2O scaling]. Cressie et al. (2016) calculated non-linear errors, using second order error analysis, and found errors on the order of 0.2 ppm, which would not fully explain the discrepancy between the predicted and true errors either in the simulation studies or real data.

Figure 2 shows the Jacobian magnitude (the $XCO_2$ Jacobian averaged over all frequencies) for $XCO_2$ versus retrieved "Band 2 albedo slope". The Jacobian for the lower (LMT) and upper (U) partial columns (described in Kulawik et al., 2017) are also plotted, and both partial columns vary the same way, e.g. same slope signs, i.e. the nonlinear interferent error would be positively correlated between the two partial columns.

The right panel of Figure 2 compares the Jacobian magnitude between matched results from configuration (c) and (d), in Table 3 for land cases with post-processing screening. The $CO_2$ Jacobian magnitude difference is up to -4% different for case (c) minus (d), and is correlated with the difference in retrieved "H2O Scaling" with correlation -0.75. Other parameters that had strong correlations (> 0.4) are: aerosol water pressure (0.55), aerosol ice pressure (0.43), surface pressure (0.41). Mapping

20 this correlation to an error in retrieved $XCO_2$ would require the calculation of second order Jacobians as in Cressie et al. (2016), and then mapping this into an error in $XCO_2$. A crude way to estimate the $XCO_2$ error resulting from these Jacobian differences is to consider the completely linear case, where radiance = K . $XCO_2$. In this case, a +1% error in the Jacobian would result in a -1% error in $XCO_2$, to fit the radiance. So the variations in the $XCO_2$ Jacobians that are seen could explain the 0.8 ppm $XCO_2$ differences from the linear estimate.

25 **3.2 Measurement error**

To validate the measurement error, results from runs with and without noise (cases (c) and (b) from Table 3) are analyzed. The standard deviation of the $XCO_2$ difference between the runs ("true error") was compared to the predicted measurement error. The two runs, which both have clouds and other interferents, as well as smoothing errors, are assumed to be identical other than one having measurement error added. The runs are compared after quality screening, which was described in Section 2.4.

Figure 3 shows the baseline and predicted measurement error. For land nadir, the average error is 0.35 ppm and the average predicted is 0.29 ppm. For ocean glint, the average error is 0.14 ppm and the average predicted error is 0.21 ppm. The bias difference between the runs with and without noise was 0.01 ppm for ocean and 0.03 ppm for land nadir.



The predicted error ranged from 0.14 to 0.70 ppm for land and 0.12 to 0.35 ppm for ocean. The correlation between the predicted error and the absolute value of the error is 0.27 for land and 0.08 for ocean, so the scene-to-scene variations in the predicted error are not very useful.

The errors for averaged observations is calculated using averaged adjacent observations. If the error for averaged observations reduces with the square root of the number of observations averaged, then the error is a random, not correlated, error. Random error is highly desirable for assimilation and other uses. For land nadir the error is shown in Table 7

10    If the error is random, then the n = 9 result should be one third the error for the n = 1 result, and this is what is found. Similarly for ocean, the error for n=9 is 1/3 of the n=1 error. The simulated data does not have the data density of actual OCO-2 data so that it is hard to say whether averaging in close proximity would behave similarly.

In summary, for these simulated cases, the measurement error is overpredicted for land by 0.06 ppm, and overpredicted for
15    ocean by 0.07 ppm, but the measurement error appears to average randomly and does not introduce a bias.

### 3.3 Smoothing error

Smoothing error represents the error introduced by imperfect sensitivity and is calculated using the averaging kernel, the true state and the prior state. The smoothing error terms from Eq. 1 are:

$$\mathbf{v}_{true\_ak} = \mathbf{v}_a + \mathbf{A}_{xx}(\mathbf{v}_{true} - \mathbf{v}_a) \qquad\qquad (7)$$

To validate the smoothing error, the non-linear retrieval is compared to the linear estimate, called v_(true_ak), and to the true. The linear estimate should compare better to the non-linear retrieval. Run (a) from Table 3 is used, which does not contain clouds in the true state (i.e., limited interferent error), and does not have measurement error in the observed radiances.

The predicted smoothing error is 0.12 ppm for ocean glint and 0.16 ppm for land nadir. Comparison between retrieved $XCO_2$ and true has a 0.0 ppm bias for ocean and 0.46 bias for land (retrieved $XCO_2$ is 0.46 ppm lower than true). The standard deviation is 0.33 ppm for land and 0.35 ppm for ocean.

30    Comparison of the retrieved $XCO_2$ versus $⟦XCO_2⟧\_true$ or $⟦XCO_2⟧\_(true\_ak)$ (true state with OCO-2 averaging kernel applied) yielded the same biases and standard deviations (within 0.02 ppm). Therefore, the use of the OCO-2 averaging kernel and prior for comparisons, using Eq. 7, does not improve the comparison quality versus OCO-2. This analysis suggests





modelers would do similarly to directly compare to OCO-2 versus applying the OCO-2 averaging kernel and prior to the model before comparing to OCO-2. However, a previous study by Wunch et al. (2011) found that for comparisons to TCCON, if the averaging kernel is not applied, it leads to 0.2 ppm seasonal biases. The current analysis shows that it does not do harm to apply Eq. 7, but that it does not help either, with the caveat that the simulated data does not cover different seasons.

**3.4 Interferent error**

Previous studies by Merrelli et al. (2015), and O'Brien et al., (2016) have found that clouds and aerosols can contribute errors larger than predicted. We look at the relationship between errors in retrieved interferents versus errors in $XCO_2$ and the prediction of the relationship as characterized by the averaging kernel.

The error in $XCO_2$ from the interferent term of Eq. 1, multiplied by the pressure weighting function, h, estimates the propagation of interferent error into $XCO_2$, shown in Eq. 8.

$$XCO_2 \text{ interferent error} = \mathbf{h}_x CO_2{}^T \mathbf{A}_{xv} (\mathbf{v}_a - \mathbf{v}_{true}) \tag{8}$$

This equation predicts that the interferent will only have an impact if the prior is different than the true, and that the impact will be proportional to the prior minus true difference, with the constant of proportionality provided by the off-diagonal averaging kernel, A_xv. Many of the interferents, e.g. H2O Scaling, parameters start at their true values for this simulation, and therefore are predicted to have no impact on $XCO_2$. Yet, large correlations in errors are seen, when comparing $XCO_2$ error versus interferent error. Taking the expected standard deviation of $XCO_2$ interferent error from Eq. 8 gives the predicted 20 interferent error, which averages 0.2 ppm for case (b) from Table 3.

We look at (retrieved minus true $XCO_2$) versus (prior minus true interferent) or (retrieved minus true interferent) in Fig. 4, using run (b) from Table 3, which has clouds but no measurement error. The red line shows hxCO2T A_xv (v_a-v_true ), the predicted relationship between the $XCO_2$ error and the prior minus true difference. For both "band 2 albedo slope", left, and 25 "H2O scaling", right, there is no predicted relationship but a strong correlation is seen.

Figure 5 shows the predicted versus true errors, including correlations. The true error is calculated from Eij =mean((retrieved minus true)i * (retrieved minus true)j) for all cases that pass post-processing quality. The true errors are much larger and show more correlations than predicted. Both matrices are normalized using the equation Eij = Eij / sqrt(E0ii*E0jj), where E is the 30 error covariance of interest and E0 is the predicted error covariance. To further analyze the interferent error, we looked at the diagonal terms of the error covariance and the correlations to $XCO_2$ in Table 8. In order for the error correlations between $XCO_2$ and interferents to be assessed, the $CO_2$ profile is mapped to $XCO_2$ using the pressure weighting function as given in





Eq xx.  Table 8 shows the predicted and true errors for all interferents, for all good quality land cases.  The error factor (EF) is calculated as:

$$EF = \sqrt{(\sigma_{true}^2 + bias_{true}^2)/\sigma_{predicted}^2} \qquad (9)$$

where the predicted standard deviations come from the predicted errors and the "true" standard deviation and bias come from the true errors.  The error factor is found to be greater than 1 for almost all parameters.

Another useful diagnostic of interferent error is the predicted error correlation between each interferent and $XCO_2$, calculated by:

$$Correlation_{ij} = error_{ij}/\sqrt{error_{ii} * error_{jj}} \qquad (10)$$

which can be compared to the actual error correlation.  Table 8 shows that for most interferents both the errors and the correlations are underpredicted.  The parameters that are both underpredicted and significantly correlated (>0.25) to $XCO_2$
errors are shown in bold.

The true effect of interferent error on $XCO_2$ can be crudely estimated by the actual slope of $XCO_2$ error (not shown in Table 8, but, the actual slope shown in Fig. 4) multiplied by the interferent error.  This estimate cannot distinguish between correlation and causation.  The standard deviation of this estimate is shown as the last column of Table 8. "Impact on $XCO_2$". The
interferent error estimated with a more simplified "surrogate" model was much smaller in Hobbs et al. (2017)

**4 Post processing bias corrections**

Post-processing analysis of real ACOS OCO-2 retrieval results has uncovered linear relationships between $XCO_2$ error and various parameters such as the retrieved surface pressure, liquid water optical depth, and $CO_2\_grad\_del$ (an estimate of the profile curvature) (Wunch et al., 2011). Similar correlations have been found between the above parameters and the lower
partial column (Kulawik et al., 2017). The standard operational procedure that has been adopted by the ACOS algorithm team for both OCO-2 and GOSAT data is to perform a bias correction of the estimated $XCO_2$ based on the linear correlations of the difference in $XCO_2$ compared to various truth metrics with certain retrieved parameters. In this section, we look specifically at the behavior of $CO_2\_grad\_del$ (defined in Section 4.1) and dP (defined in Section 4.2) bias correction in the simulated system. The purpose of the analysis of this section is to answer the following questions:
(1)     Do the bias correction for dP and $CO_2\_grad\_del$ behave similarly in the simulation system as in real OCO-2 retrievals?
     (2)     What is the effect of bias correction on $CO_2$ errors?





### 4.1 CO2_grad_delta

CO2_grad_del is defined as delta[20] – delta[13] where delta is the retrieved $CO_2$ profile minus the prior $CO_2$ profile, [20] is the surface level, and [13] is 7 levels above the surface, i.e., 0.63*(surface pressure). CO2_grad_delta represents curvature of

the retrieved $CO_2$ profile that differs from the prior. It has been found that the slope of $XCO_2$ error versus CO2_grad_delta varies depending on the a priori covariance that is used in the retrieval system, with a more evenly varying covariance having less dependency of $XCO_2$ error versus CO2_grad_delta (O'Dell, unpublished result). The standard OCO-2 constraint is very loose at the surface (e.g. with 50 ppm a priori variability) and tighter in the mid-Troposphere (with ~10 ppm a priori variability). Most $CO_2$ variability does occur near the surface near the primary sources and sinks, but the apriori constraint used in the

retrieval algorithm would favor variations at the surface even in cases when the variations occur at a higher level due to the weighting due to the prior covariance.

Figure 6 shows errors in $XCO_2$, LMT, and U partial columns versus CO2_grad_delta for configuration (b). In the simulated retrievals, the values of the slope of delta $xCO_2$ versus CO2_grad_delta is -0.001 and -0.008 for land and ocean, respectively.

It is clear that there are significant errors in the partitioning between the lower (LMT) and upper (U) partial columns that are correlated to CO2_grad_delta. The slope of LMT versus CO2_grad_delta is 0.23 and 0.22 for land and ocean, respectively and -0.07 and -0.08 for for U land and ocean, respectively. For real ACOS-GOSAT (b3.5) data, Kulawik et al. (2017) found a slope of 0.39 for land and 0.31 for ocean for LMT and -0.11 and -0.09 for U land and ocean, respectively, which are similar values as seen in this simulated data.

These results naturally lead to the question; what is the effect of placing $CO_2$ at the wrong pressure level? The mean Jacobian for the U partial column (upper 15 layers) is only about 60% (0.62) of the mean value for the lowermost 4 layers. Therefore a molecule in the LMT partial column is equivalent to about 1.6 molecules in the upper partial column. Therefore, a molecule mistakenly placed in the lower 4 layers and moved to the upper layers in the post-processing step needs to be exchanged for

1.6 molecules in the upper partial column to have the same impact on the radiances at the new level. At CO2_grad_delta of 35, for land, LMT is high by ~8.4 ppm. For an even exchange, moving 8.4 ppm from the LMT partial column to the U partial column results in +2.5 ppm in the U partial column ONLY from the effects of air mass (because the U partial column has more air mass; = 8.4 ppm *.23 LMT airmass / 0.77 U airmass). Considering the difference in sensitivity, and multiplying by 1.6, this corresponds to +4.0 ppm in the U partial column. The net effect on $XCO_2$ of this bias correction is the sum of the partial

columns times the air mass, -8.4*.23 + 4.0*.77 = 1.1 ppm. This is at CO2_grad_delta of 35, so that would mean that the slope for $XCO_2$ error versus CO2_grad_delta is 0.031. For real OCO-2 v7 data, the slope of $XCO_2$ error versus CO2_grad_delta is +0.0280 and -0.077 for land, ocean, respectively (Mandrake et al., 2017). This analysis explains a positive slope in $XCO_2$



versus CO2_grad_delta; but would not explain a negative slope. The negative slope would result from additional correlations or errors acting in addition to this effect.

**4.2 dP**

The quantity dP is the difference between retrieved and prior surface pressure and is used as a post-processing bias correction

for OCO-2. In this section, we explore results from dP in the simulated dataset to try to understand why bias correction based on this parameter is useful.

Although it is typically assumed that the surface pressure is determined solely from the O2A band, the strong and weak $CO_2$ bands also contribute.  , For land nadir, averaged over cases passing post-processing quality screening, the band-averaged

Jacobian strengths in the weak and strong $CO_2$ bands relative to the O2A band are 0.2 and 0.4, respectively. Based on the surface pressure Jacobian and the spectral error, a value of -2 hPa will create a spectral bias 0.2 times the size of the spectral error in the O2A band, which, because it is a correlated error, will be an additive error over the band.

Figure 7 shows the actual error covariances and biases for 3 different subsets of run (d):  dP < -2 hPa, -1<dP<1hPa ("nominal

cases"), and dP>1.5hPa. The errors shown are normalized by the predicted error, using the equation $Eij = Eij / sqrt(E0ii*E0jj)$, where E is the error covariance of interest and E0 is the predicted error covariance. A diagonal value of 1 means that the actual error is the same as predicted, and a diagonal value of 4 represents an actual error that is twice (sqrt(4)) as large as predicted. The errors and error correlations are much larger than predicted for many parameters. In addition, the $CO_2$ parameters show less correlation with other parameters for the nominal case.  Also note that the nominal case has less saturation, meaning less

errors and correlations.

Next we looked at the possibility of screening incorrect surface pressure results using the reduced $\chi^2$ ($\chi^2$  normalized by the expected noise).  To do this we used the $\chi^2$ for land, cloudy cases with dP<-2 vs. -1<dP<0.  The cases with dP<-2 had 0.04, 0.01, and 0.06 higher reduced $\chi^2$ in the 3 bands, respectively.  Although the dP<-2 case fit the spectra worse there was too

much overlap to distinguish between these cases solely from the reduced $\chi^2$.

The albedo errors and correlations (purple box) particularly stand out, with correlations with many retrieved parameters.  The albedo terms are, in order:  O2A mean, O2A slope, weak mean, weak slope, strong mean, strong slope.  Based on the O2A mean albedo and the surface pressure Jacobians, a change in retrieved surface pressure of -2 hPa can be compensated by a

change in the albedo on the order of -0.001, with this analysis based on band averages, and not necessarily implying a good fit. However, this analysis indicates that very minute changes in the surface albedo (on the order of 0.1%) can compensate for moderate sized errors in the retrieved surface pressure.  The exact relationship can be better studied by examining the radiative





transfer, and looking at how the final transmission of sunlight relates to both the total amount of atmospheric absorption and the surface albedo.

Error in the retrieved $XCO_2$, lower partial column (lmt) and upper partial columns (u) are plotted versus the error in surface pressure in Fig. 8, which all show moderate (R=0.63) to strong (R=-0.98) correlations. For ocean soundings, the OCO-2 v7 dP correction factor is -0.08, while the simulated data has a slope of +0.15. For land scenes the OCO-2 v7 dp correction factor is -0.3, and the correction factor for this simulated data is -0.23. Note that for the simulated data, the prior surface pressure is set to the true, so (surface pressure – prior) is the same as (surface pressure – true). The bias correction factors are found in Table 4 of the v7 bias correction documentation.

The retrieval system must match the mean photon path length for the O2A channel using retrieved parameters like surface pressure, albedo, water, temperature, aerosol pressure heights, and aerosol optical depths. Also note that the number of O2 molecules is fixed and not retrieved. Mean photon path length increases with higher albedo and aerosol optical depth (Palmer et al., 2001). Additionally, moving aerosols lower in the atmosphere increases mean photon path length, because light scattered by the aerosol travels farther, and a larger surface pressure will increase mean photon path length because the path length to
the surface is longer. The retrieval system varies these parameters to match the observed radiances. Ideally, the 3 bands would have the same albedo and aerosol properties, so that getting the O2A band mean photon path length right will also get the mean photon path length in the $CO_2$ bands. Real aerosol optical depths tend to be higher in the O2A band then in the $CO_2$ bands. However, the aerosol optical depth versus frequency is fixed for OCO-2. Therefore, as an example, using a too-thick
aerosol in the O2A band to compensate for a too-small surface pressure will not balance in the $CO_2$ bands because the same too-small surface pressure will be offset by less aerosol. The relative strengths of the Jacobians for the four aerosol optical depths in the O2A versus $CO_2$ bands are 1.5x, 3.3x, 7.2x, and 2.1x, respectively, indicating the dominance of the O2A-band concerning aerosol information.

As seen in Fig. 7b, for dp < -2 hPa, there is a negative bias in surface pressure (because we selected for this), negative biases in 3 of the 4 aerosol optical depths (green box, parameters 1, 4, and 10), positive bias in retrieved aerosol pressure (green box, parameters 2, 5, 8), and negative biases in the retrieved albedo (purple box, parameters 1, 3, 5). The error covariances show that within this subset of observations, there are also strong negative correlations between retrieved surface pressure error and errors in albedo and errors in aerosol optical depth and positive correlations between error in aerosol optical depth and errors
in albedo.

To trace the interferent errors to an error for $XCO_2$ the effect of each bias on mean photon path length for the O2A, weak and strong bands needs to be calculated and then the mean photon path length error of the $CO_2$ bands versus the O2A band will give the error for $XCO_2$. For example if the O2A mean photon path length is perfect and the $CO_2$ mean photon path length is





0.5% too large relative to true, then the $CO_2$ retrieved VMR will be 0.5% too small. Since aerosols are compensating for errors in surface pressure, it is not ideal to fix their relationship versus frequency.

Figure 7d-f shows the bias patterns for these different groups. Comparing 7d,e, and f, reveals patterns that could be used for
screening: e.g. low bias in Kahn1 aerosol optical depth and a low biases in all albedo means and high biases in all albedo slope indicates a negative surface pressure error; whereas a high bias in Kahn1 aerosol pressure and width and a high bias in the strong band albedo slope indicates a positive surface pressure error. In real retrievals, since true is not known; e.g. an albedo high bias versus an actual high albedo cannot be distinguished; however a particular pattern of biases versus the priors would be suspicious.

It is interesting to note that the system appears to be able to compensate and pass post-processing quality screening, using albedo and aerosols, for low surface pressure biases down to -4 hPa, but high surface pressure biases only up to +2 hPa.

**4.3 Error correlation and effect of bias correction on errors**

Another important question is; how does bias correction within the $CO_2$ column affect errors, particularly the error correlations
in XCO$_2$ and the partial columns? Kulawik et al. (2017) found that the predicted error correlation between the LMT and U partial columns was -0.7 for land and -0.8 for ocean; whereas the actual error correlation versus aircraft was found to be +0.6 (with uncertainty in the correlation due to the fact that aircraft do not cover the full U partial column and effects of co-location error). Additionally, Kulawik et al. (2017) found that whereas the XCO$_2$ predicted errors were underestimated by about a factor of 2, the LMT and U errors were overestimated by about a factor of 2. Weakening the LMT and U correlations would
result in higher and more accurate error estimates for XCO$_2$.

The errors for XCO$_2$, LMT, and U for land and ocean for configuration (b) are summarized in Table 9. The bias correction for XCO$_2$ (using only CO2_grad_del and dP) lowers the XCO$_2$ bias from 0.2 to 0.1 ppm and the error from 0.8 ppm to 0.7 ppm for land, but has no impact on the ocean error or bias. The XCO$_2$ error is underestimated by a factor of 2 for these simulation
results, similarly to what was found with real data.

Similar to findings with real data, the XCO$_2$ error in these simulations is underestimated, whereas the LMT and U errors are overestimated. However, the overestimate of the partial column errors are not as large as seen with real GOSAT data. The predicted error correlation is -0.91 for the LMT and U errors, whereas the actual error correlation is -0.5. Using eq. 10c from
Kulawik et al. (2017), and the LMT and U errors in Table 9, we note two key results. First, the XCO$_2$ predicted error is 0.37 ppm when the error correlation is -0.91. Second, the predicted XCO$_2$ error is 0.64 (0.71) ppm for ocean (land) when the actual correlation is -0.57 (-0.46) for ocean (land). The second result is close to the actual error of 0.7 ppm. The estimate of +0.6



correlation from Kulawik et al. (2017) is probably wrong, and could be due to unaccounted effects of co-location error on correlation estimates.

As seen in Section 3.1, non-linearities from interferents affect both partial columns similarly. This would result in positive error correlation (since the correlation is strongly negative and results in a less negative correlation than predicted) and explain the larger actual versus predicted $XCO_2$ error. A high negative correlation is desirable for $XCO_2$ because it asserts that, although there is uncertainty in the partitioning of LMT and U, the sum of the two has a smaller uncertainty.

## 6 Discussion and conclusions

The 15 orbits of simulated retrievals result in ~10,000 land and ocean scenes for cloud-free runs, and 870 and 680 land and ocean cases for runs with clouds, after post-processing quality screening. Prior to application of quality flags, described in Section 2.3, the errors are ~10 ppm for land and ~2 ppm for ocean. After quality flags, and bias correction are applied, the errors are 0.7 ppm, with mean bias errors of 0.1 ppm for both land and ocean.

Comparing runs with and without measurement noise added to the radiances showed that the predicted measurement error is accurate. Comparing the retrieved results to the linear estimate using only the $CO_2$ parameters ("smoothing error") shows that the smoothing error is not greater than 0.5 ppm, but due to interferent error and non-linearity this could not be validated more accurately with the tests used. A more accurate way to validate this would be to run tests with different priors (e.g. Kulawik et al., 2008), which was previously done (unpublished) finding that the smoothing errors are smaller than 0.2 ppm.

The linear estimate does not predict the non-linear retrievals to better than 0.9 ppm (much worse when quality flags are not used), indicating this level of non-linearity in the retrieval system. The interferent error is underpredicted by a factor of 4, based on the relationship of $XCO_2$ error versus error for each retrieved interferent. The retrieved interferent error is twice as large as predicted for some parameters, and the correlation between the retrieved interferent error and $XCO_2$ error is twice as large as predicted for some parameters. The larger correlation is likely due to the fact that $CO_2$ Jacobian strength is correlated with many retrieved interferent values; a wrong interferent value will result in the wrong $CO_2$ Jacobian strength, resulting in an error in $CO_2$.

Two bias correction terms are explored: CO2_grad_delta, the gradient of the retrieved $CO_2$ profile relative to the priori; and dP, the retrieved surface pressure minus the prior. The CO2_grad_delta bias correction could be explained by 1) a loose $CO_2$ constraint near the surface prefers changes near the surface versus changes elsewhere. 2) Since the $CO_2$ Jacobian strength near the surface is stronger versus the Jacobian elsewhere in the profile, molecules incorrectly placed near surface are underestimated, because each molecule has "too much" effect on the observed radiance, 3) this results in an $XCO_2$ column that




is too low. This explanation would explain the positive bias correction factor seen in OCO-2 v7 land and v8 land and ocean, but would not explain the negative correction factor seen in v7 ocean.

The theoretical basis for dP is complicated because so many other retrieval parameter errors are correlated to errors in dP. This makes sense from a fundamental radiative transfer perspective which ties together the surface and scattering properties with the amount of atmospheric column for any particular sounding. The retrieval system appears to use albedo and aerosols to compensate for errors in dP. In these simulated results the dP bias correction has a similar slope as seen in real OCO-2 data for land, but not for ocean. The results with dP errors had marginally higher radiance residuals but not high enough to easily screen.

Similar to the findings in Kulawik et al. (2017), the $XCO_2$ column error is much higher than predicted, whereas the lower and upper partial $CO_2$ column errors, LMT and U, respectively, have errors lower than predicted. The underprediction of $XCO_2$ error results because the retrieval system thinks the LMT and U partial column error correlation is -0.91. The actual correlation is -0.5 to -0.6 after bias correction, with the uncorrected results having both higher error and higher correlations in the partial

columns. When the actual correlation is used to estimate $XCO_2$ error, the predicted $XCO_2$ error matches the actual error within 0.1 ppm. The reason why this correlation is off may be due to the fact that both partial column Jacobian strengths vary similarly with interferent errors, which are underpredicted in the linear estimates of errors, and would result in less negative correlation between the partial columns.

These results suggest a few possible strategies (a) isolating the primary interferent parameters via pre-retrievals of aerosols with surface pressure, $CO_2$, and albedo fixed, followed by a full joint retrieval. This would allow clouds and aerosols to be approximately set without throwing the other retrieved parameters off. A similar technique was employed in the thermal infrared to mitigate cloud contamination (e.g. Eldering et al., 2008). A second tactic would be to perform retrievals beginning at many different initial states, selecting the result with the lowest radiance residual. This solution however is computationally

expensive.

In summary, the simulated retrievals have many of the same attributes of real data, with the advantage of knowledge of the true state and ability to see what is going on under the hood. These simulation studies suggest attention should be given to non-linearity, because the ability to estimate errors and make incremental improvements depends on the accuracy of the linear

estimate, which has accuracy of only about 0.9 ppm in these simulation studies.

**Author contributions:**



Susan Kulawik set the direction of the research and did the analysis, figures, and was the primary manuscript writer. Rob Nelson and Tommy Taylor generated simulated OCO-2 true states, radiances, and retrievals. Chris O'Dell guided the work of the simulation system and advised on the analysis. All authors participated in the manuscript writing and editing.

5 **Competing interests:**

The authors declare that they have no conflict of interest.

**Acknowledgements:**

This work is supported by NASA Roses proposal, "Assessing OCO-2 predicted sensitivity and errors", 10 NMO710771/NNN13D771T. Plots are made using matplotlib (Hunter, 2007).





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



Table 1.  Retrieved parameters in this simulation study

| Index | Parameter |
|---|---|
| 1-20 | 20 $CO_2$ volume mixing ratios at 20 pressure levels from the surface to top of the atmosphere (20) |
| 21 | Water vapor scaling factor |
| 22 | Surface pressure |
| 23 | Temperature profile offset |
| 24, 27, 30, 33 | Aerosol optical depth for 4 types |
| 25, 28, 31, 34 | Aerosol pressure height for 4 types |
| 26, 29, 32, 35 | Aerosol width for 4 types (prior uncertainty is very tight) |
| 36, 38, 40 | Albedo mean for 3 bands |
| 37, 39, 41 | Albedo slope for 3 bands |
| 42, 43, 44 | Dispersion offset for 3 bands (frequency offset). |
| 45 | Wind speed (ocean).  In the original files this is index 36, but was moved to index 45 so that the albedo indices are consistent between land and ocean. |
| 45, 46 | Fluorescence (Land).  The true fluorescence is set to zero for these simulations. |



Table 2.  Updates in the simulated retrieval system since O'Dell et al. (2012)

| B2.10 changes | B3.3 changes | B3.4 changes |
|---|---|---|
| 1) Gaussian aerosol profiles | 1) Residual fitting | 1) Ocean surface parameterization |
| 2) Sigma pressure levels | 2) Reduced p_surf prior uncertainty | 2) Update weak $CO_2$ spectral range |
| 3) Update to prior $CO_2$ profile | 3) Prior AOD=0.05 | 3) Spectroscopy update |
| 4) Spectroscopy updates | 4) Spectroscopy update | |
| 5) Correction to $XCO_2$ AK | 5) Fluorescence fit land gain H (GOSAT) | |



Table 3. Configurations used in this work.

| Case | Measurement Error | Clouds+Aerosols | Comment |
|------|-------------------|-----------------|---------|
| (a) | No | No | Smoothing only |
| (b) | No | Yes | Smoothing + interferent |
| (c) | Yes | Yes | Smoothing + interferent + measurement error |
| (d) | No | Yes | Different water prior/initial |



Table 4. Number of cases for each configuration. The "clouds in true==yes" cases contain many fewer soundings than "no clouds" because of pre-screening. The #good is from post-processing screening

|  | Clouds in true | # | # good (post-screening) | configuration (from Table 3) |
|---|---|---|---|---|
| Land (nadir) | No | 12,097 | 10,229 | a |
| Ocean (glint) | No | 14,265 | 11,468 | a |
| Land (nadir) | Yes | 3,445 | 868/869/768 | c/b/d |
| Ocean (glint) | Yes | 1,560 | 679/674/620 | c/b/d |





Table 5. Mean bias and standard deviation between retrieved and true, sorted by standard deviation. The bold entries are the nominal cases most closely simulating actual OCO-2 retrievals.

| Case from Table 3 | Land/Ocean | Clouds in true | Post-processing screening | Meas. error | Bias | Standard deviation |
|---|---|---|---|---|---|---|
| (a) | Ocean | No | Yes | No | -0.1 | 0.4 |
| (a) | Land | No | Yes | No | -0.5 | 0.4 |
| (a) | Ocean | No | No | No | -0.3 | 0.6 |
| (a) | Land | No | No | No | -0.5 | 0.7 |
| (b) | Ocean | Yes | Yes | No | 0.1 | 0.7 |
| **(c)** | **Ocean** | **Yes** | **Yes** | **Yes** | **0.1** | **0.7** |
| (b) | Land | Yes | Yes | No | 0.2 | 0.8 |
| **(c)** | **Land** | **Yes** | **Yes** | **Yes** | **0.2** | **0.8** |
| (b) | Ocean | Yes | No | No | -0.6 | 2.7 |
| (b) | Land | Yes | No | No | -2.3 | 10.3 |



Table 6. Difference of linear estimate versus non-linear retrieval, noise-free, cloud, quality-screened cases

|  | Land bias | Land std | Ocean bias | Ocean std |
|---|---|---|---|---|
| Predicted | 0 | 0.3 | 0 | 0.2 |
| Retrieved vs. $CO_2$ AK | -0.2 | 0.8 | -0.2 | 0.6 |
| Retrieved vs. full AK | -0.4 | 0.8 | -0.9 | 0.8 |
| Retrieved vs true | 0.2 | 0.9 | 0.1 | 0.7 |



Table 7. Error versus averaging for measurement error

| n (number averaged) | Error land (ppm) | Error ocean (ppm) |
|---|---|---|
| 1 | 0.35 | 0.14 |
| 2 | 0.25 | 0.10 |
| 3 | 0.20 | 0.08 |
| 9 | 0.12 | 0.05 |



Table 8. Predicted and actual errors for interferents and correlations between interferents and XCO$_2$ for simulated land retrievals for case (b) from Table 3. Bold values are those parameters with interferent errors larger than predicted and large actual correlations to XCO$_2$ error (absolute value larger than 0.25).

| | | Pred error | Actual error | Error factor | Pred corr | Actual corr | Impact on XCO$_2$ (ppm) |
|---|---|---|---|---|---|---|---|
| **Met** | **H2O scaling** | **0.003** | **0.005 ± 0.004** | **5** | **0.35** | **0.93** | **1.2** |
| Met | Surface pressure | 0.5 | -0.67 ± 1.02 | 3 | -0.38 | -0.02 | 0.0 |
| **Met** | **Temperature offset** | **0.04** | **0.25 ± 0.22** | **9** | **0.17** | **0.44** | **0.6** |
| **Aerosol** | **Aerosol ice OD** | **0.002** | **-0.02 ± 0.21** | **101** | **0.03** | **0.81** | **1.1** |
| Aerosol | Aerosol Ice Pressure | 0.09 | 0.03 ± 0.28 | 3 | -0.01 | 0.22 | 0.3 |
| Aerosol | Aerosol Ice Width | 0.01 | 0.01 ± 0.01 | 2 | -0.00 | 0.13 | 0.2 |
| **Aerosol** | **Aerosol Kahn1 OD** | **0.01** | **-5.0 ± 0.8** | **4** | **-0.36** | **-0.39** | **0.5** |
| Aerosol | Aerosol Kahn1 Pressure | 0.3 | 0.3 ± 0.4 | 1 | 0.08 | -0.10 | 0.1 |
| Aerosol | Aerosol Kahn1 Width | 0.01 | 0.04 ± 0.08 | 9 | -0.00 | -0.19 | 0.3 |
| Aerosol | Aerosol Kahn2 OD | 0.01 | -5.0 ± 0.8 | 2 | 0.32 | -0.02 | 0.0 |
| **Aerosol** | **Aerosol Kahn2 Pressure** | **0.4** | **0.7 ± 0.5** | **2** | **-0.02** | **0.26** | **0.3** |
| Aerosol | Aerosol Kahn2 Width | 0.01 | 0.1 ± 0.09 | 17 | 0.00 | 0.18 | 0.2 |
| Aerosol | Aerosol Water OD | 0.008 | -5.9 ± 1.0 | 7 | -0.06 | -0.15 | 0.2 |
| Aerosol | Aerosol Water Pressure | 0.4 | 0.7 ± 0.5 | 3 | -0.01 | -0.06 | 0.1 |
| Aerosol | Aerosol Water Width | 0.01 | 0.09 ± 0.03 | 9 | 0.00 | -0.13 | 0.2 |
| **Albedo** | **Band 1 Albedo ave** | **0.0008** | **-0.0002 ± 0.003** | **3** | **0.19** | **-0.50** | **0.7** |
| Albedo | Band 1 Albedo slope | 1e-6 | 1e-6 ± 1e-6 | 3 | -0.26 | -0.10 | 0.1 |
| **Albedo** | **Band 2 Albedo ave** | **0.0006** | **-0.002 ± 0.004** | **7** | **0.19** | **-0.54** | **0.7** |
| Albedo | Band 2 Albedo slope | 1e-7 | 2e-6 ± 2e-6 | 8 | 0.10 | 0.20 | 0.3 |
| Albedo | Band 3 Albedo ave | 0.0007 | -0.001 ± 0.005 | 7 | 0.04 | -0.22 | 0.3 |
| **Albedo** | **Band 3 Albedo slope** | **1e-6** | **0e-6 ± 2e-6** | **3** | **0.14** | **-0.36** | **0.5** |





Table 9. Predicted and actual errors and biases in raw and bias-corrected simulated data run with configuration (b) from Table 3. Similar to operational retrievals, bias-corrected $XCO_2$ error is underestimated, whereas the $CO_2$ partial column errors are overestimated. The $XCO_2$ error underprediction results from overestimated error correlations of the partial columns.

| | Ocean pred. (ppm) | Ocean actual (ppm) | Ocean actual corrected (ppm) | Land pred. (ppm) | Land actual (ppm) | Land actual corrected (ppm) |
|---|---|---|---|---|---|---|
| LMT | 2.6 | 2.6 ± 2.9 | 0.1 ± 2.3 | 3.3 | 2.9 ± 4.2 | -0.6 ± 2.6 |
| U | 0.9 | -0.6 ± 1.1 | 0.2 ± 1.0 | 1.2 | -0.6 ± 1.4 | 0.3 ± 0.9 |
| $XCO_2$ | 0.3 | 0.1 ± 0.7 | 0.1 ± 0.7 | 0.4 | 0.2 ± 0.8 | 0.1 ± 0.7 |
| LMT and U correlation | -0.91 | -0.67 | -0.57 | -0.90 | -0.68 | -0.46 |



**Figure captions:**

Figure 1. Scatter plots of $XCO_2$ difference from the prior for retrieved versus true on the simulated data. This corresponds to dataset (c) with clouds and measurement error, and post-processing screening applied for land (left) and ocean (right), with

1:1 plots shown on the top panels, and histogram of the differences on the lower panels.

Figure 2. $XCO_2$ (black), lower $CO_2$ partial column (red), and upper $CO_2$ partial column (blue) Jacobian band-averaged magnitude versus interferent parameters. Left shows $CO_2$ magnitude versus retrieved "Band 2 Albedo slope", using configuration (b) from Table 3; right shows the $CO_2$ Jacobian magnitude difference (in percent) for matched cases from run

(b) and (d) versus differences in retrieved "$H_2O$ scaling".

Figure 3. Histogram of difference between $XCO_2$ with noise on and noise off for ocean(left) and land(right), cases (b) and (c) from Table 3.

Figure 4. Predicted (red line) and true error (red dots) for two interferents, "Band 2 albedo slope", left, and "H2O Scaling", right.

Figure 5. Predicted and true errors. Left shows the predicted error covariance matrix, for the retrieval parameters listed in Table 1, with the $CO_2$ profile collapsed into 2 parameters [LMT and U partial columns]. The blue, orange, green, and purple

boxes contain $CO_2$, metrological, aerosol, and albedo parameters, respectively. Both matrices are normalized by the diagonal of the predicted errors.

Figure 6. Error in retrieved $CO_2$ for $XCO_2$ (black), upper partial column, U (blue) and lower partial column LMT (red) versus CO2_grad_delta for ocean (left) and land (right)

Figure 7. Normalized actual error covariances and biases of retrieved parameters for dp<-2 hPa (a, d), -1<dp<1 hPa (b, e), and dp > 1.5 hPa (c, f) using configuration from Table 3 (d) for land/cloudy. The purple box surrounds the albedo parameters, the green box surrounds aerosol parameters, the red box surrounds metrological parameters, and the blue box surrounds the $CO_2$ fields, which have been collapsed into lower and upper partial columns. The errors are normalized by the predicted errors

(which are shown in Fig. 5). The arrow in panel (a) shows correlation between LMT and surface Pressure, which is negative (also see Fig. 8b below)



Figure 8. Error in the lower partial column (LMT), upper partial column (U) and total column (XCO$_2$) versus error in surface pressure (with 0.2 hPa bins) for ocean (left) and land (right). The OCO-2 v7 XCO$_2$ bias versus dP is -0.3 for land and -0.08 for ocean.



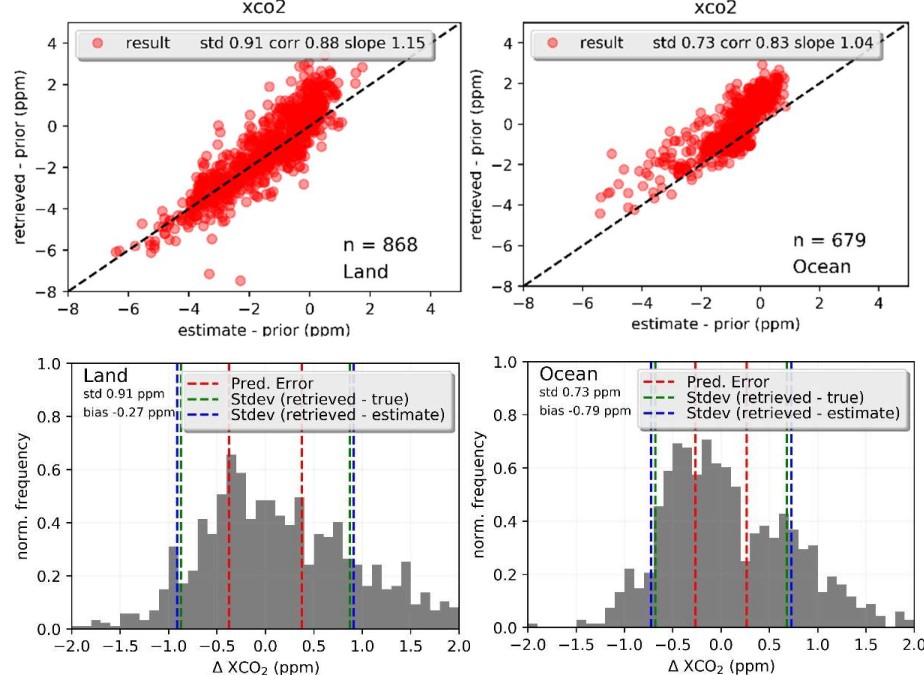

Figure 1. Scatter plots of $XCO_2$ difference from the prior for retrieved versus true on the simulated data. This corresponds to

5  dataset (c) with clouds and measurement error, and post-processing screening applied for land (left) and ocean (right), with

1:1 plots shown on the top panels, and histogram of the differences on the lower panels.





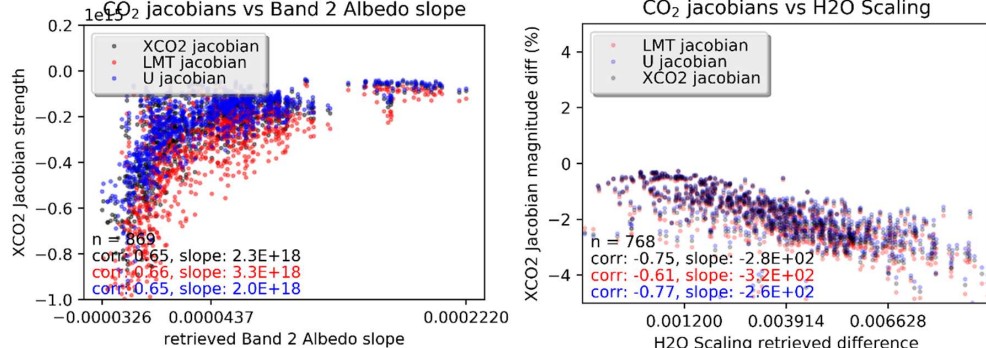

Figure 2. $XCO_2$ (black), lower $CO_2$ partial column (red), and upper $CO_2$ partial column (blue) Jacobian band-averaged magnitude versus interferent parameters. Left shows $CO_2$ magnitude versus retrieved "Band 2 Albedo slope", using configuration (b) from Table 3; right shows the $CO_2$ Jacobian magnitude difference (in percent) for matched cases from run 5 (b) and (d) versus differences in retrieved "$H_2O$ scaling".



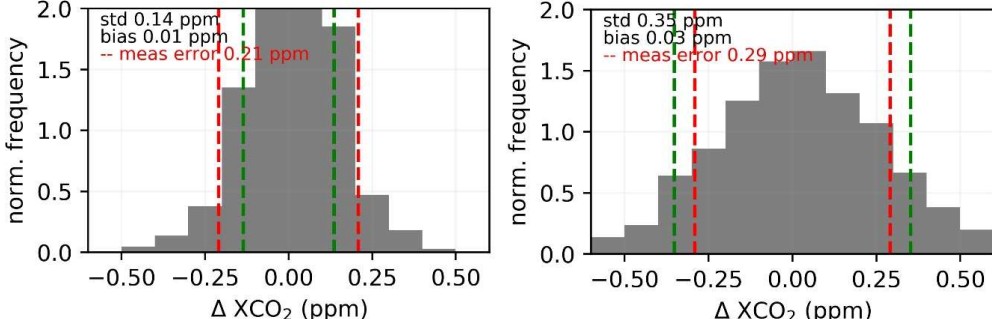

Figure 3. Histogram of difference between $XCO_2$ with noise on and noise off for ocean(left) and land(right), cases (b) and (c) from Table 3.





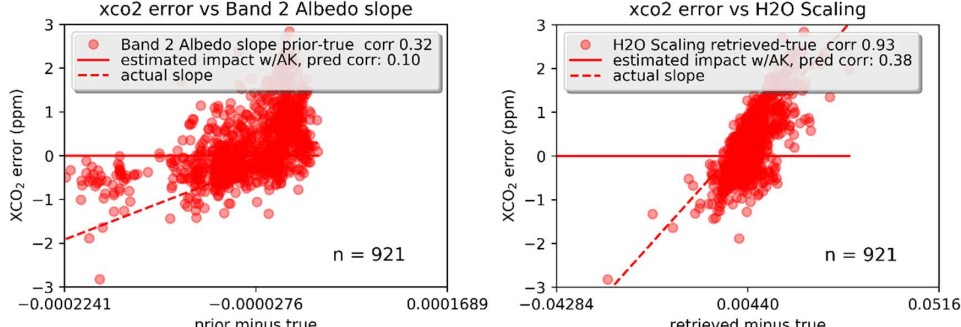

Figure 4. Predicted (red line) and true error (red dots) for two interferents, "Band 2 albedo slope", left, and "H2O Scaling",

right.





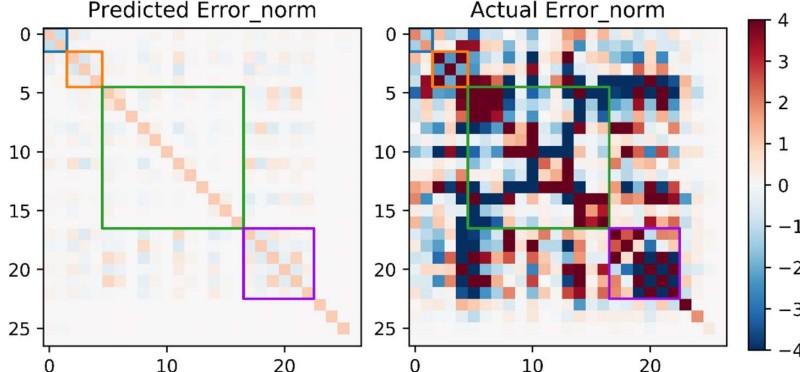

Figure 5. Predicted and true errors. Left shows the predicted error covariance matrix, for the retrieval parameters listed in Table 1, with the $CO_2$ profile collapsed into 2 parameters [LMT and U partial columns]. The blue, orange, green, and purple boxes contain $CO_2$, metrological, aerosol, and albedo parameters, respectively. Both matrices are normalized by the diagonal of the predicted errors.



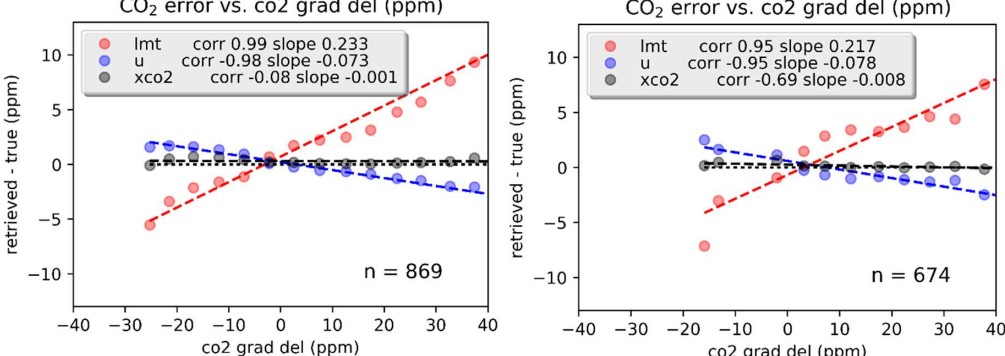

Figure 6. Error in retrieved $CO_2$ for $XCO_2$ (black), upper partial column, U (blue) and lower partial column LMT (red) versus
CO2_grad_delta for ocean (left) and land (right)





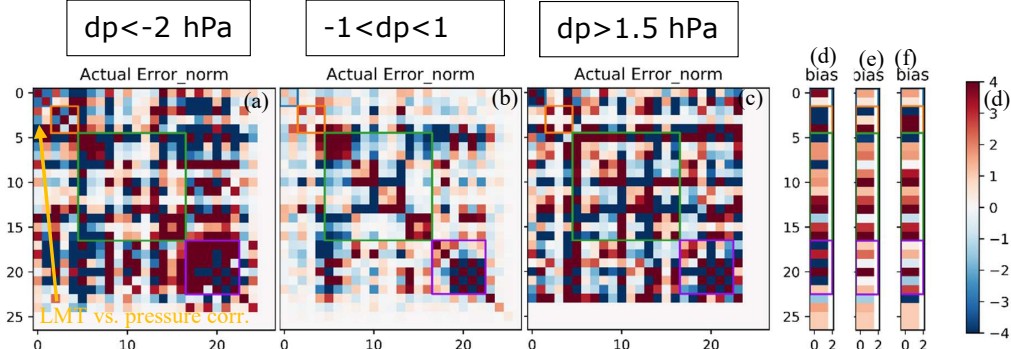

Figure 7. Normalized actual error covariances and biases of retrieved parameters for dp<-2 hPa (a, d), -1<dp<1 hPa (b, e), and
dp > 1.5 hPa (c, f) using configuration from Table 3 (d) for land/cloudy. The purple box surrounds the albedo parameters, the
green box surrounds aerosol parameters, the red box surrounds metrological parameters, and the blue box surrounds the $CO_2$
fields, which have been collapsed into lower and upper partial columns. The errors are normalized by the predicted errors
(which are shown in Fig. 5). The arrow in panel (a) shows correlation between LMT and surface Pressure, which is negative
(also see Fig. 8b below)





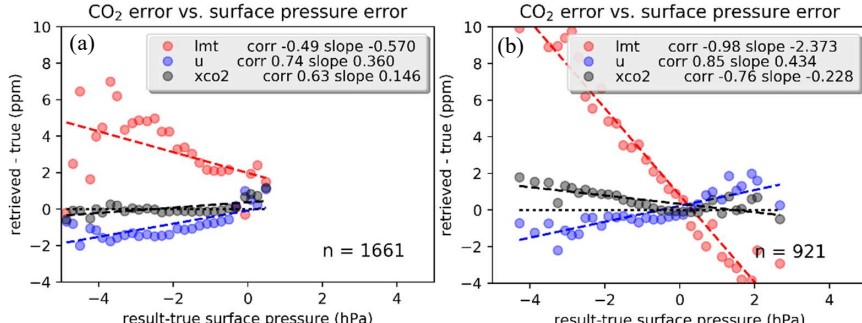

Figure 8. Error in the lower partial column (LMT), upper partial column (U) and total column ($XCO_2$) versus error in surface pressure (with 0.2 hPa bins) for ocean (left) and land (right). The OCO-2 v7 $XCO_2$ bias versus dP is -0.3 for land and -0.08 for ocean.

