# Peer review of "Validation of OCO-2 error analysis using simulated retrievals"

_Atmospheric Measurement Techniques, 2018_

## Referee Comment (RC1) · Anonymous Referee #1 · 7 Feb 2019

The paper by Kulawik and colleagues "Validation of OCO-2 error analysis using simulated retrievals" is a thorough piece of work which advances the state of the art. AMT is the ideal journal for this study. I recommend publication after fixing a number of mostly minor issues. I have spotted only one major issue (see below). At many instances there seem to be formatting errors, just as if LaTex maths commands were used outside the maths environment or in a non-LateX document, or similar. This makes reading unnecessarily difficult and causes the impression that the paper has been finished in a hurry. Given the inherent scientific quality of the paper, it deserves a more careful presentation.

Major issue:

p6 l5, Eq 6: This is not the $\chi^2$ of an optimal estimate because the constraint term is missing. Without the additional term $(\hat{x} - x_a)^T S_a^{-1}(\hat{x} - x_a)$ I think the expectation value will not be the number of degrees of freedom of the retrieval system (See Rodgers book Eq. 2.43).

Minor and technical issues:

p1 l14: what about adding "... larger than predicted *by linear error estimation*,..."

p1 l25: shouldn't it read "was launched"

p2 l1: either "analysis follows"" or "analyses follow"

p3 l16: "...linearity of the retrieval system *in the vicinity of the result*..." Actually it is not assumed that the retrieval system is linear but that it is only moderately nonlinear (in Rodgers' language). That is to say, that the system behaves approximately linear in $\pm 1\sigma$ around the result.

p3 l18: I thought you talk about error estimation but here you talk about the retrieval. Please clarify. Perhaps "Error estimation based on retrievals using a..."?

p4 l10 "...the a priori *covariance matrix for* $CO_2$ *has the dimension*..."

p4 l12 I might have missed something but it is not clear to me what 'aircraft variability'

is. I am not sure if the term 'error' is adequate in the context of a priori uncertainty.

p4 l14 I suggest to add the term 'assumed' somewhere. Either "The assumed a priori errors" or "are all assumed uncorrelated".

p4 l35 "Sainv" have you pasted a LaTeX macro into a word document here?

p5 l 33 and throughout: I suggest to avoid these technical abbreviations like "CO2_grad_delta" in the text as far as possible and to use common language instead. If you do not want to use common language for these terms, then please replace the variable name of the computer code by a variable in mathematical notation.

p6 l1 this should read $\chi^2$; there are numerous errors of this type. I do not mention each single one.

p6 l5 try to avoid computer language type variable names, replace by mathematical notation.

p7 l20 'errors due to physics that is perfectly described by the retrieval forward model' not quite clear what is meant. Please reword.

p11 l17. I do not think that the smoothing error describes the error introduced by the 'imperfect sensitivity'. Imperfect sensitivity will cause retrieval noise. The cause of the smoothing error is that $x_{true} \neq x_a$.

p8 l19 should this read "second moment"?

[Figure]

Subsection headers 4.1 and 4.2: please avoid variable names in the subsection headers.

p14 l13: Not sure if abbreviation 'LMT' has been defined. I might have missed the definition but please check.

p15 l9: There is something wrong after '...contribute.'

p15 l22: I thought that the reduced $\chi^2$ means $\chi^2$ divided by the degrees of freedom. Isn't normalization with the related inverse covariance matrix inherent in the $\chi^2$ by definition?

---

## Referee Comment (RC2) · Sourish Basu (Referee) · 29 Mar 2019

The manuscript by Kulawik et al investigates possible error sources in OCO2 L2 retrievals by the ACOS algorithm in an OSSE setting, with the caveat that not all the error sources in the real retrievals have been characterized. This is a careful study that merits publication in AMT, after the authors have responded to my (mostly minor) comments and suggestions. Overall, the manuscript would benefit from some copy editing; I've pointed out such errors where they confuse the message, but I cannot be certain that there are not more.

1. *Page 2, line 8:* Typo or possible missing words in "finds that non-linear retrievals this relatively simple simulation".

2. *Page 2, line 21:* Vague antecedent in "these simulated results", do "these" refer to Connor et al (2016) or the current work?

3. *Page 2, lines 26-27, and elsewhere:* Suggest making double quotes consistent throughout the document. Currently they're a mixture of "quotes" (preferred), "quotes" and "quotes".

4. *Page 3, line 7:* Typo, "retrievel"

5. *Page 3, line 30:* The claim that the performance of systems (3) and (4) were comparable is a strong one, since (4) includes a lot of complicating effects not in (3). It seems that the authors compared the two systems to arrive at this conclusion, the "preliminary studies" referred to here. I would like to see some sort of evidence from those studies, i.e., why do they think that the performances are comparable? This is not just idle curiosity; the authors themselves say that their error estimates are larger than earlier estimates by Hobbs et al (2017) using a surrogate model, which raises the question of whether choice (3) indeed is sufficient to capture most of the error sources.

6. *Page 4, line 35:* Typo, $Sainv \rightarrow S_a^{-1}$

7. *Page 5, equation (5), and page 12, equation (8):* Typo, $h_x CO_2^T \rightarrow h_{XCO_2}^T$ or something like that, right now it looks like $CO_2$ is the vector that's being transposed

8. *Page 6, lines 26-30:* It was not clear to me whether the current work used the newer scheme (which picks the two most likely aerosol types per scene) or the older one.

9. *Page 7, lines 1-5:* The authors downsample from 24 soundings per second to 1 sounding per second. While I understand this choice from the point of computational convenience, this has the potential for changing inter-sounding corre-

lations, and whether errors average down over multiple soundings (e.g., the top part of page 11). Can the authors comment? Would the conclusions in the top half of page 11 still hold for real OCO2 retrievals?

10. *Page 7, lines 16-17:* Why is a realistic cloud screener necessary for this work, given that coverage is not the focus of this investigation? Interfering errors from clouds are important, of course, but cloud screening to throw out soundings prior to retrieval should not affect the conclusions of this work, right?

11. *Page 7, line 26:* Define "true" retrieval errors before this sentence. Currently it's defined on line 33.

12. *Page 8, lines 18-19:* Do the biases in table 5 average down with the number of soundings? Or are they true biases that are independent of the number of sounding used to calculate them (with variations due to finite sample size)?

13. *Page 9:* Define the linear estimate and how it's calculated before discussing it. For calculating the linear estimate from equation (1), are the Jacobians/averaging kernels evaluated at the prior state vector values or the posterior values from the nonlinear solution?

14. *Page 9, line 9:* Are the 1.3 and 1.0 ppm figures biases or standard deviations (random errors)?

15. *Page 11, line 27:* I'm surprised by the 0.0 ppm bias, is this because there are no clouds in the true state for this exercise?

16. *Page 12, lines 1-4:* Seasonality of the effect of the averaging kernel is one reason for applying it to models, another is the possibility of spatial patterns. The data in this study do not span multiple seasons, but it does span multiple surface types, albedos, aerosol loading, etc., all of which influence the averaging kernel. Does the impact of applying the averaging kernel and prior have a spatial pattern?

[Figure]

17. *Page 12, line 25:* What does it mean that there is no predicted relationship but a strong correlation? Does it mean that the correlation is arising because both variables are impacted by some common element in the state vector?

18. *Page 13, line 1:* Unresolved reference to "Eq xx".

19. *Page 14, line 4:* "gradient", not "curvature"

20. *Page 16, paragraph 2:* Here and elsewhere, it is not clear to me how a bias correction is done in this OSSE setup. For real OCO2 retrievals, the retrieved $XCO_2$ are compared to any of a set of truth metrics, and linear relationships derived between the errors and co-retrived parameters. In the OSSE, what supplies the truth metric? Just the "true" state that is already known (because this is an OSSE)? In that case, is the bias correction formula applied derived specifically for this OSSE, or is the formula for real v7 retrievals used? It would seem to be more appropriate to use the former, but lines 8-9 here suggest that the latter was used. Why is that valid?

21. *Page 16, lines 12-13:* I did not know that the ACOS algorithm kept the number of $O_2$ molecules fixed. How is this done, is it computed from the surface pressure and explicitly kept fixed? In that case, how does the surface pressure change during the retrieval? Purely due to water? And if so, is this change in water (which leads to $dP \neq 0$) consistent with the water column in the retrieval?

---

## Author Comment (AC1) · 14 Jun 2019

Thank you to reviewer 1 for the helpful comments. We have responded to all comments.

Reviewer 1 major comment: p6 l5, Eq 6: This is not the 2 of an optimal estimate because the constraint term is missing. Without the additional term $(\hat{x}..x_a)^T S..1 a (\hat{x}..x_a)$ I think the expectation value will not be the number of degrees of freedom of the retrieval system (See Rodgers book Eq. 2.43).

Response: Thank you for pointing this out. As the reviewer notes, the (retrieved minus prior) contributes to the chi2 used for the goodness-of-fit in an optimal estimate. For oco-2 fits, there four "bands" contributing to the chi2, the O2A, the weak, the strong, and the state deviation from the prior. However, the purpose of Eq. 6, is to see how

well each spectral band is fit for different categories. e.g. see Section 4.2 dP, where the spectral fit is checked for different surface pressure error ranges. This diagnostic was also renamed, "rad_chiˆ2" so as to avoid confusion with the chi2 used in the retrieval. This section was reworded to explain better the purpose of this diagnostic, "One useful diagnostic is an estimate of how well the modeled radiances match the observed radiances for each of the three OCO-2 spectral bands."

The same terminology was updeated in 2.3 (quality flags) and 4.2 (dP) to be consistent.

p1 l14: what about adding "... larger than predicted by linear error estimation,..." Response: Updated as suggested.

p1 l25: shouldn't it read "was launched" Response: Updated as suggested.

p2 l1: either "analysis follows" or "analyses follow" Response: Wording changed to, "OCO-2 error analysis uses Rodgers (2000)..."

p3 l16:"...linearity of the retrieval system in the vicinity of the result..." Actually it is not assumed that the retrieval system is linear but that it is only moderately nonlinear (in Rodgers' language). That is to say, that the system behaves approximately linear in 1 around the result. Response: Thank you for noting this. Updating to "1) Linear estimates of errors, which assumes moderate linearity of the retrieval system"

p3 l18: I thought you talk about error estimation but here you talk about the retrieval. Please clarify. Perhaps"Error estimation based on retrievals using a..."? Response: Updated 2), 3), and 4) as suggested, e.g. "2) Error estimates from non-linear retrievals of simulated radiances using a fast, simplified radiative transfer, called the surrogate model (Hobbs et al., 2017)."

p4 l10 "...the a priori covariance matrix for CO2 has the dimension..." Response: Updated the wording as suggested.

p4 l12 I might have missed something but it is not clear to me what "aircraft variability" is. I am not sure if the term "error" is adequate in the context of a priori uncertainty.

Response: The variability of the aircraft is outside the scope of this paper. Wording updated to, "The larger variability near the surface allows more variability in the retrieved CO2 profile near the surface."

p4 l14 I suggest to add the term "assumed" somewhere. Either"The assumed a priori errors" or, "are all assumed uncorrelated". Response: Wording updated as suggested, to "The a priori errors for other parameters are all uncorrelated in the a priori covariance... "

p4 l35"Sainv" have you pasted a LaTeX macro into a word document here? Response: It looks like the word file got mangled in transit between co-authors, and have updated many similar issues in the text.

p5 l 33 and throughout: I suggest to avoid these technical abbreviations like "CO2_grad_delta" in the text as far as possible and to use common language instead. If you do not want to use common language for these terms, then please replace the variable name of the computer code by a variable in mathematical notation. Response: Updated this to a mathematical notation which I cannot paste into this text box, but first introduced near the end of the abstract.

p6 l1 this should read 2; there are numerous errors of this type. I do not mention each single one. Response: This notation was updated in response to the major comment, above. The notation was also fixed.

p6 l5 try to avoid computer language type variable names, replace by mathematical notation. Response: Updated to r, for radiance, in Eq. 6.

p7 l20 "errors due to physics that is perfectly described by the retrieval forward model" not quite clear what is meant. Please reword. Response: This was worded confusingly. Updated wording to, "This error analysis ideally would use the exact same forward model in both the L1b simulations and the L2 retrieval algorithm, as our analysis assumes that Eq. 1 should be valid, without errors from forward model differences."

p11 l17. I do not think that the smoothing error describes the error introduced by the "imperfect sensitivity". Imperfect sensitivity will cause retrieval noise. The cause of the smoothing error is that xtrue != xa. Response: I see how imperfect sensitivity results in noise which is not smoothing error. Updated wording to, "Smoothing error occurs when the averaging kernel deviates from the identity matrix..."

p8 l19 should this read, "second moment"? Response: Yes, thank you.

Subsection headers 4.1 and 4.2: please avoid variable names in the subsection headers. Response: Updated names to "The retrieved profile gradient" and "The retrieved surface pressure".

p14 l13: Not sure if abbreviation "LMT" has been defined. I might have missed the definition but please check. Response: Yes they were defined earlier, but now are re-referenced Section 2.1 and the paper Kulawik et al. (2017) here.

p15 l9: There is something wrong after "...contribute." Response: Cleaned up extra punctuation, line now reads, "Although it is typically assumed that the surface pressure is determined solely from the O2A band, the strong and weak CO2 bands also contribute information."

p15 l22: I thought that the reduced 2 means 2 divided by the degrees of freedom. Isn't normalization with the related inverse covariance matrix inherent in the 2 by definition? Response: This notation and definition was cleaned up when Eq. 6 was tidied. It is no longer called "reduced" but uses Eq. 6 directly.

---

## Author Comment (AC2) · 14 Jun 2019

Thank you to reviewer 2 for the helpful comments. We have responded to all comments.

Reviewer 2 responses:

1. Page 2, line 8: Typo or possible missing words in "finds that non-linear retrievals this relatively simple simulation". Response: Added missing word, "However this study finds that non-linear retrievals using this relatively simple simulation..."

2. Page 2, line 21: Vague antecedent in "these simulated results", do "these" refer to Connor et al (2016) or the current work? Response: It refers to the current work. Updated wording, "The linear analysis of Connor et al. (2016) does not explain the higher errors in this work, because the simulations in this work do not include unaccounted

errors sources."

3. Page 2, lines 26-27, and elsewhere: Suggest making double quotes consistent throughout the document. Currently they're a mixture of "quotes" (preferred), "quotes" and "quotes". Response: Changed all to "" quotes.

4. Page 3, line 7: Typo, "retrievel" Response: Updated and did spell check throughout document.

5. Page 3, line 30: The claim that the performance of systems (3) and (4) were comparable is a strong one, since (4) includes a lot of complicating effects not in (3). It seems that the authors compared the two systems to arrive at this conclusion, the "preliminary studies" referred to here. I would like to see some sort of evidence from those studies, i.e., why do they think that the performances are comparable? This is not just idle curiosity; the authors themselves say that their error estimates are larger than earlier estimates by Hobbs et al (2017) using a surrogate model, which raises the question of whether choice (3) indeed is sufficient to capture most of the error sources. Response: The initial runs using the more complicated system have been lost, and the statement that (3) and (4) are comparable is not supported by any analysis in this paper. This statement was made weaker, "...because preliminary studies seemed to find that the performance of systems (3) and (4) were comparable (results not shown)" Regardless of whether (3) and (4) are comparable, this paper uses system (3). This paper, using system (3) finds comparable errors to those found with the actual OCO-2 system, and a similar ratio of 2 between actual and predicted errors. We think that the fact that our errors are larger than Hobbs et al (2017) (which uses the surrogate model, indicates that the simplified radiative transfer used in Hobbs et al (2017) does not result in realistic errors.

6. Page 4, line 35: Typo, Sainv ! S..1a Response: Updated this notation. Also, checked other equations for similar errors.

7. Page 5, equation (5), and page 12, equation (8): Typo, hxCOT 2 ! hT XCO2 or

something like that, right now it looks like CO2 is the vector that's being transposed
Response: Agree this was mangled. Updated this here and one other location.

8. Page 6, lines 26-30: It was not clear to me whether the current work used the newer scheme (which picks the two most likely aerosol types per scene) or the older one. Response: This work uses the scheme where the same 4 aerosols are always selected. Updated wording, "In the older L2 algorithm versions (pre B3.5), also used in this work... "

9. Page 7, lines 1-5: The authors downsample from 24 soundings per second to 1 sounding per second. While I understand this choice from the point of computational convenience, this has the potential for changing inter-sounding correlations, and whether errors average down over multiple soundings (e.g., the top part of page 11). Can the authors comment? Would the conclusions in the top half of page 11 still hold for real OCO2 retrievals? Response: The measurement error, with this dataset, was random and reduced as 1/sqrt(N) in this study. The question as to whether it would average for denser data, likely it would. The caveat was weakened to "The simulated data does not have the data density of actual OCO-2 data so while averaging in close proximity would likely behave similarly, there is some uncertainty."

10. Page 7, lines 16-17: Why is a realistic cloud screener necessary for this work, given that coverage is not the focus of this investigation? Interfering errors from clouds are important, of course, but cloud screening to throw out soundings prior to retrieval should not affect the conclusions of this work, right? Response: This work processes pre-generated simulated radiances consistently to how real data is processed. Part of the process is cloud screening, which is never 100% accurate, sometimes letting through cloudy cases and sometimes screening out clear cases. After retrievals, quality flags are applied, and can result in some true cloudy cases being flagged as good cases without clouds. So, it is important to test the end-to-end system, and not only give the system non-cloudy cases. Added text to better explain this to the reader. "It is important to test the system from end-to-end with radiances containing a variety of

cloud conditions, because the cloud screening is never 100% accurate, sometimes letting through cloudy cases, and because quality flags can sometimes flag cloudy cases being as good quality without clouds. "

11. Page 7, line 26: Define "true" retrieval errors before this sentence. Currently it's defined on line 33. Response: Updated wording to define exactly what we are comparing, "Our goal in this work is to compare linearly predicted vs. actual errors in XCO2..."

12. Page 8, lines 18-19: Do the biases in table 5 average down with the number of soundings? Or are they true biases that are independent of the number of sounding used to calculate them (with variations due to finite sample size)? Response: Table 5 column heading was updated to "Mean bias". This is the bias over all the data and does not average away. A paragraph and figure were added to Section 2.4 to look at the spatial distribution of biases. The new paragraph is:

"Correlated biased errors are seen in real OCO-2 data, with correlations in time, e.g. ~60 days (Kulawik et al., 2019), at small spatial scales, e.g. < 1 degree (Worden et al., 2016), and at medium spatial scales, e.g. 5-10 degrees (Kulawik et al., 2019). Although this dataset cannot probe a seasonally dependent bias, as it covers only 1 day of observations, it can be used to probe spatial patterns of the biases. However, note that probing very small spatial patterns will be difficult to see because of the small amount of data processed in comparison to real OCO-2. A plot showing the spatial pattern of retrieved minus true is shown in Fig. 2 panel (a), which shows a high bias near the equator and a low bias near the poles. Panel (b) of Fig. 2 shows the difference between true XCO2 and XCO2 with the OCO-2 averaging kernel. The overall spatial pattern in panel (a) is not affected by the application of the averaging kernel, which makes sense because the averaging kernel effect is ~0.2 ppm whereas the differences are on the order of 0.9 ppm. An analysis of the correlation scale length of (retrieved minus true) XCO2 finds a correlated error of 0.3 ppm and full-width half-maximum in the bias of ~3 degrees, which is similar to the correlated error of 0.4 ppm and scale

length of ∼5-10 degrees found in Kulawik et al., 2019. The simulated data has accurate meteorology (temperature, winds, etc.) that drive the simulated true states, but the cloud and aerosol spatial structures are not very accurate, so that the spatial scales are not expected to be identical between this simulated dataset and real OCO-2 data. This analysis shows that correlated biases exist in simulated data, and that simulated data is useful for exploring the characteristics and even more importantly, the cause of regional biases."

13. Page 9: Define the linear estimate and how it's calculated before discussing it. For calculating the linear estimate from equation (1), are the Jacobians/averaging kernels evaluated at the prior state vector values or the posterior values from the nonlinear solution? Response: The linear estimate was previously introduced in Section 2.1 but not referred back in the text on page 9. Text added in Section 2.1 to better introduce the linear estimate, "The linear estimate describes the response of the retrieval system to instrument errors and incorrect a priori inputs, based on the strengths of the Jacobians (representing sensitivity of the radiances to the retrieval state) and constraints (how much pressure is applied to parameters to stay near the a priori inputs). The linear estimate in Eq. 1 is used to estimate the errors, and for simulations, where we know all the inputs, it is useful to test each component of Eq 1." Page 9 now has the updated text, "To test the system linearity the linear estimate, using Eq. 1, and discussed in Section 2.1 is compared to the non-linear retrieval result. The inputs to Eq. 1 include the instrument noise (if on), a priori covariance, and Jacobians at the final retrieved state."

14. Page 9, line 9: Are the 1.3 and 1.0 ppm figures biases or standard deviations (random errors)? Response: These are the single-observation errors. I updated to v8 error estimates from Kulawik et al., 2019 (in prep). The text was updated to, "For real OCO-2 v8 data, comparisons to TCCON for single-observation land nadir and ocean glint show errors (including both random and systematic errors) of 1.0 and 0.8 ppm, respectively (Kulawik et al., 2019), meaning that the real errors are comparable

to these simulated data errors. Real OCO-2 data has location-dependent biases on the order of 0.5-0.6 ppm (Wunch et al., 2017; Kulawik et al., 2019)..."

15. Page 11, line 27: I'm surprised by the 0.0 ppm bias, is this because there are no clouds in the true state for this exercise? Response: The 0.0 ppm is the mean bias. There is still a spatial pattern to the bias, listed as the standard deviation. The spatial pattern of true_ak minus true is also now shown in Fig. 2 and a discussion was added about the spatial distribution of biases in Section 2.4.

16. Page 12, lines 1-4: Seasonality of the effect of the averaging kernel is one reason for applying it to models, another is the possibility of spatial patterns. The data in this study do not span multiple seasons, but it does span multiple surface types, albedos, aerosol loading, etc., all of which influence the averaging kernel. Does the impact of applying the averaging kernel and prior have a spatial pattern? C3 Response: Yes, the analysis of the spatial pattern of the bias was previously lacking in the paper. The spatial pattern of the difference between the retrieved and true or retrieved and true with the averaging kernel applied were very similar. The application of the averaging kernel did not affect the spatial pattern. This makes sense because the AK application is a 0.2 ppm effect, and the systematic error is ~0.6 ppm (Kulawik et al., 2019). The analysis of the spatial pattern of the differences between retrieved and true, or true_ak and true was added to Section 2.4 as described above in the answer to #12.

17. Page 12, line 25: What does it mean that there is no predicted relationship but a strong correlation? Does it mean that the correlation is arising because both variables are impacted by some common element in the state vector? Response: We have found that the $CO_2$ Jacobian strength varies with the retrieved water. If the retrieved water is not the true water it will result in the wrong strength $CO_2$ Jacobian. This will affect the retrieved $CO_2$ value but will not be predicted to have an effect. A sentence was added to this section, "This could be explained by the results from Section 3.1, showing that the XCO2 Jacobian strength varies with the retrieved albedo or retrieved water, whereas the error analysis assumes that the Jacobian strength does not vary."

18. Page 13, line 1: Unresolved reference to "Eq xx". Response: Fixed this, refers to Eq. 5.

19. Page 14, line 4: "gradient", not "curvature" Response: Updated wording to gradient.

20. Page 16, paragraph 2: Here and elsewhere, it is not clear to me how a bias correction is done in this OSSE setup. For real OCO2 retrievals, the retrieved XCO2 are compared to any of a set of truth metrics, and linear relationships derived between the errors and co-retrived parameters. In the OSSE, what supplies the truth metric? Just the "true" state that is already known (because this is an OSSE)? In that case, is the bias correction formula applied derived specifically for this OSSE, or is the formula for real v7 retrievals used? It would seem to be more appropriate to use the former, but lines 8-9 here suggest that the latter was used. Why is that valid? Response: This was not worded well. We calculated the bias correction for this simulated dataset. We compare the bias correction on this simulated dataset to the bias correction found in v7. We don't expect the exactly same relationships, but we would expect similarities, assuming that the biases are caused by the effects studied in this simulated dataset. Added wording in Section 4, "The bias correction is determined using this simulated dataset, and then applied to the same dataset, which is somewhat circular, since the true is both used to determine the bias correction and to check the bias correction, but it is important to know whether the relationships exist. For example, what causes the spatial patterns seen in the bias in Fig. 2. " Clarified the wording in section 4.2, "The bias found in this work for this simulated dataset for the XCO2 bias versus dP is -0.23 for land and 0.15 for ocean. We can compare these to the OCO-2 v7 biases of -0.3 for land and -0.08 for ocean. "

21. Page 16, lines 12-13: I did not know that the ACOS algorithm kept the number of O2 molecules fixed. How is this done, is it computed from the surface pressure and explicitly kept fixed? In that case, how does the surface pressure change during the retrieval? Purely due to water? And if so, is this change in water (which leads to dP 6=

0) consistent with the water column in the retrieval? Response: This was not worded well. The ACOS algorithm keeps the O2 volume mixing ratio fixed not the number of O2 molecules. The retrieved surface pressure then affects the number of molecules. Updated wording to, "Also note that the O2 volume mixing ratio (VMR) is fixed and not retrieved."
* * *
[Figure]

[Figure]

Figure 2. (a) Spatial pattern of XCO₂ retrieved minus true for case (b) from Table 3 (cloudy but no measurement error), with quality screening applied. Panel (b) shows the difference between true XCO₂ with the OCO-2 averaging kernel applied minus true XCO₂.

**Fig. 1.** Added new Figure 2 showing spatial pattern of bias in simulated data